# Incidence of dry eye symptoms and behavioural-cultural risk factors among university students population in Jordan

May M. Bakkar[1]*, Mona Aridi[2], Mohammad A. Alebrahim[1], Wissam Ghach[3,4]*

1 Faculty of Applied Medical Sciences, Jordan University of Science and Technology, Irbid, Jordan, 2 University of Angers, LARIS, SFR MATHSTIC, F-49000 Angers, 62 avenue Notre Dame du Lac, Angers, France, 3 Department of Public Health, Canadian University Dubai, Dubai, United Arab Emirates, 4 The Global Health Network – Middle East and North Africa (TGHN-MENA), Dubai, United Arab Emirates

* wissam.ghach@cud.ac.ae (WG); mmbakkar@just.edu.jo (MMB)

## Abstract

### Purpose

To estimate the incidence of dry eye (DE) symptoms among university students in Jordan and to examine the relationship between behavioral and cultural risk factors and DE symptom severity.

### Methods

A cross-sectional study involving 788 university students was conducted in Jordan. Participants' mean age was 21.87 years (SD = 3.824; range: 18–45 years). The incidence and severity of DE symptom were assessed using the validated Arabic version of the Ocular Surface Disease Index (ARB-OSDI) questionnaire, administered through Google Forms. The survey included demographic questions and behavioral-cultural risk factors (smoking and eye cosmetic use). One-Way ANOVA and multi-regression analyses were used to investigate the association between OSDI mean scores and behavioral-cultural risk factors.

### Results

The incidence of DE symptoms, defined as an OSDI score ≥ 13, was 74.2% among university students. Higher DE symptom severity was statistically associated with females' gender (p < 0.001), older age (≥27 years) (p = 0.032), contact lens use (p = 0.001), frequent use of eye cosmetics (p < 0.001), and a history of DED (p < 0.001). Smoking habits, including the use of Dokha or Ajami, smoking in enclosed spaces, and daily smoking, were also associated with increased DE symptom severity (all p < 0.001). Contributing factors to the high incidence and severity of DE symptom included long-term use of eye cosmetics (particularly mascara and internal eyeliner) and sleeping while wearing contact lenses.

**Data availability statement:** All relevant data are within the paper and its Supporting Information files.

**Funding:** Research Grant No: 20230271 from Deanship of Research at Jordan University of Science and Technology.

**Competing interests:** The authors have declared that no competing interests exist.

## Conclusion

Dry eye symptoms are highly prevalent among university students in Jordan and are significantly associated with factors such as age, gender, contact lens use, cosmetic application, and tobacco consumption.

## 1. Introduction

Dry eye disease (DED) is a highly prevalent and multifactorial condition commonly encountered in eye clinics and widely reported by patients across the globe. The Tear Film and Ocular Surface Society-Dry Eye Workshop II (TFOS DEWS II) defines DED as a "multifactorial disease of the ocular surface characterized by a loss of homeostasis of the tear film and accompanied by ocular symptoms, in which tear film instability and hyperosmolarity, ocular surface inflammation and damage, and neurosensory abnormalities play etiological roles" [1].

Various tools and methodologies have been developed to estimate the incidence of DED both clinically and symptomatically.

The global incidence of DED ranges from 5% to 50%, influenced by geographical region, demographics, and the diagnostic criteria used in studies [2]. Arab countries report the highest incidence rates [3]. In Jordan, the incidence of symptomatic DED was reported by Bakkar et al. to be as high as 59% (4), based on a study that assessed DE symptoms only using a non-validated version of the OSDI questionnaire and did not include clinical diagnostic tests [4].

Smoking is a significant global public health issue, with tobacco linked to, tear film abnormalities, ocular surface defect and retinal nerve fiber layer damage [5–7]. In Jordan, smoking remains prevalent, with 40.2% of Jordanians smoking [8], including a high incidence among university students (23%–35%) who use conventional cigarettes and waterpipes [9]. However, limited studies have explored the association between smoking behaviors, tobacco types, and DED incidence in Jordan [10].

The exposure to irritating toxicants (e.g., eye cosmetics) may also lead to discomfort, allergic reactions, and ocular changes [11]. These products are known to affect lipid layer quality and play a main role in the initiation and/or exacerbation of DED and its related symptoms [12]. However, limited studies investigated the relationship between cosmetic use and the incidence of DE symptoms in Jordan where females are expected to be predominantly applying cosmetic products [10].

To the best of our knowledge, the incidence of DE symptoms and its association to the behavioral-cultural risk factors has not been investigated among the university students in Jordan. For the first time, this paper aims to estimate the incidence of DE symptoms using ARB-OSDI tool and evaluate its statistical association to the tobacco and cosmetic use among the university students in Jordan. The expected outcome is to bridge the awareness gap among university students through community-based interventions and awareness campaigns. Local government authorities, non-governmental organizations, and eye health professionals will use the study findings

as current evidence to develop new research and educational initiatives aimed at raising awareness of DE symptoms and their behavioral and cultural risk factors.

## 2. Material and methods

A cross-sectional study was conducted between April 2023 and December 2023 to estimate the incidence of DE symptoms and its associated risk factors among a population of university students in Jordan. Using a convenience sampling technique, a google survey was distributed throughout social media platforms, and research departments in the Jordanian universities.

A formal sample size calculation was not conducted prior to the study due to the use of a convenience sampling approach aimed at recruiting as many participants as possible across all Jordanian governorates. Nonetheless, the achieved sample size of 788 respondents provided a sufficiently large dataset for statistical analysis and generalizable findings within the study's scope.

To further verify the adequacy of the sample size, a post hoc power analysis was conducted using G*Power (version 3.1.9.7) for the One-way ANOVA tests employed in the study. Power estimates were calculated for variables with four groups (smoking rate, smoking areas, and all the cosmetics variables) and 6 groups (smoking type), which represent the range of groupings used in the analysis. For the variable with four groups, the analysis yielded a non-centrality parameter ($\lambda$) = 49.25, numerator df = 3, denominator df = 784, critical F = 2.616, and a power of 0.999995. For the variable with six groups, the results were similarly robust: $\lambda$ = 49.25, numerator df = 5, denominator df = 782, critical F = 2.226, and power = 0.999977. These results confirm that the sample size was statistically robust and sufficiently powered to detect moderate effects across the ANOVA models used in the study.

### 2.1. Study population

Based on the geographical location of the respondents, the study population included 788 university students from the twelve Governorates of Jordan. The population was then stratified into groups based on gender, age, place of residence per Governorate, field of study, study level, studying duration (hours per day), use of electronic device (hours per day), contact lens wear, history of DE symptoms. Exclusion criteria include youngsters aged less than 18 years old; non-university students; and university students with eye surgeries, active ocular diseases, and ocular or systemic medications that are known to interfere with tear film production or ocular surface integrity.

### 2.2. Study tool

The validated Arabic version of the OSDI questionnaire; ARB-OSDI questionnaire was utilized in the current study to assess dry eye symptoms induced by environmental factors over the preceding week [13].

The OSDI questionnaire was originally developed by the Outcomes Research Group at Allergan Inc. (Irvine, California) and designed to provide a quick assessment of symptoms related to eye irritation and DED [14]. The 12-item questionnaire is composed of three sections: five ocular symptom items, four vision-related function items, and three environmental trigger items. Each item was scored on a scale of 0–4, where 0 indicates none of the time; 1, some of the time; 2, half of the time; 3, most of the time; and 4, all of the time. The total OSDI score was then calculated according to the following formula:

$$OSDI\ Score = \frac{Sum\ of\ scores\ for\ all\ questions\ answered\ \times 100}{Total\ number\ of\ questions\ answered\ \times 4}$$

Following the calculation of total OSDI score, OSDI status was stratified and classified with a cut-off value of score ≥ 13 into three intervals: normal [0,12], mild-to-moderate [13,32], and the interval of severe cases [33,100] [15,16].

Behavioral-cultural risk factors was assessed by types of smoked tobacco (regular cigarette, electronic cigarette, medwakh pipe, mouassal waterpipe - "flavored" and "honeyed" paste of tobacco, tumbak/ajami waterpipe - "non-honeyed" pure dark paste of tobacco); rate of smoking (daily, weekly, and monthly); indoor-outdoor areas of smoking; quantification of cigarettes, waterpipes, or medwakh pipes per week along with the smoking duration (in years); kind of eye cosmetic (mascara, eyeliner, eyeshadow, and eyelashes); duration of eye cosmetic use (< 6 months, 6–12 months, and > 12 months) along with its daily application (in hours); cleaning methods (water, soapy water, cleansing products) along with the rate of night cleansing (never, sometimes, always); and sleep routine (with/without wearing lenses and/or eye cosmetic).

## 2.3. Data analysis

The data were analyzed using the Statistical Package for Social Sciences, version 21 (SPSS, International Business Machine Corp. IBM, Chicago, IL, USA). The incidence of DE symptoms was represented by % frequency of OSDI score ≥ 13 (mild, moderate, and severe OSDI status). A one-way ANOVA was conducted to compare mean differences in the outcome variable (OSDI score) across groups stratified by independent variables such as electronic device usage hours, studying hours, and smoking patterns. The ANOVA test assessed the statistical significance of group differences, with F-statistics and p-values guiding the interpretation of results.

To investigate the linear relationship between the outcome variable (OSDI score) and multiple independent variables, a stepwise multiple regression analysis was performed. Initially, the predictors included electronic device usage hours per day, studying hours per day, smoking years, number of cigarettes smoked per week, number of Dokha smoked per week, and number of Shisha smoked per week. The stepwise method systematically retained predictors with significant contributions to the model, yielding a final regression equation that included electronic device usage hours per day, studying hours per day, and smoking years as the retained variables.

The model's goodness of fit was assessed using the correlation coefficient (R), coefficient of determination ($R^2$), and adjusted $R^2$. Statistical significance was determined through the F-statistics from the ANOVA table of the regression model. To understand the interplay of variables, control for confounding effects, and highlight significant predictors in a regression model, Zero-order correlation, partial correlation, and part correlation were interpreted.

All data analyses were conducted at a significance level of 0.05, with $P < 0.05$ indicating statistical significance and confidence intervals set at 95%.

## 2.4. Ethical considerations

The Institutional Review Board (IRB) at Jordan University of Science and Technology (JUST) has reviewed and approved the protocol of the study (2023/95). All the respondents have been informed that confidentiality and potential risk were controlled during data collection and analysis as stated in the written informed consent (first page of the questionnaire).

## 3. Results

A total of 788 university students participated in this study. The majority were female (56.7%) and aged between 18 and 23 years (81.6%). Participants' mean age was 21.87 years (SD = 3.824; range: 18–45 years). Of the participants' field and year of study, 39% were enrolled in medical, pharmaceutical, or health-related programs, and 88.5% were undergraduate students. Most participants were non-smokers (64.8%) and did not wear contact lenses (92.2%). Additionally, 61.3% of the participants had not been clinically diagnosed with DED, (Table 1).

### 3.1 Dry eye symptoms and participants' profiles

The Ocular Surface Disease Index (OSDI) assessment indicated that 74.2% of participants showed symptoms of DED. As seen in Table 1, females, older-aged group (≥ 27 years old), CL wearers, and participants who clinically diagnosed with DED exhibited the highest severity of symptoms, as reflected by higher OSDI mean scores, with statistically significant

**Table 1. Distribution of frequency, relative percentages, OSDI mean score, and One-way ANOVA with respect to gender, age group, field of study, study level and CL usage, eye drops use, history of DED, and ocular surface disease index (OSDI) status.**

| Variable | | Frequency (%) | Mean OSDI (SD) | One-Way ANOVA |
|---|---|---|---|---|
| Gender | Male | 341 (43.3%) | 24.27 (20.60) | F = 40.45 **P < 0.001** |
| | Female | 447 (56.7%) | 34.38 (23.17) | |
| Age Interval | 18-20 | 365 (46.3%) | 29.16 (21.73) | F = 3.71 **P = 0.011** |
| | 21-23 | 278 (35.3%) | 30.13 (22.72) | |
| | 24-26 | 65 (8.2%) | 25.45 (22.07) | |
| | ≥ 27 | 80 (10.2%) | 37.11 (25.62) | |
| Field of Study | Natural Sciences | 73 (9.3%) | 30.28 (21.59) | F = 1.57 P = 0.140 |
| | Health Sciences | 77 (9.8%) | 25.19 (18.78) | |
| | Medical and Pharmaceutical Sciences | 230 (29.2%) | 29.93 (23.34) | |
| | Engineering Sciences | 145 (18.4%) | 31.03 (23.02) | |
| | Art and Media Sciences | 35 (4.4%) | 31.55 (20.63) | |
| | Business and Tourism | 70 (8.9%) | 36.22 (23.71) | |
| | Education and Social Sciences | 77 (9.8%) | 26.62 (21.67) | |
| | Humanities and Political Sciences | 81 (10.3%) | 29.89 (24.36) | |
| Level of Study | Bachelor (1st year) | 166 (21.1%) | 29.17 (22.63) | F = 1.35 P = 0.232 |
| | Bachelor (2nd year) | 215 (27.3%) | 27.85 (20.53) | |
| | Bachelor (3rd year) | 154 (19.5%) | 31.55 (24.02) | |
| | Bachelor (4th year) | 162 (20.6%) | 29.44 (21.22) | |
| | Master (1st year) or equivalent | 36 (4.6%) | 34.55 (24.59) | |
| | Master (2nd year) or equivalent | 44 (5.6%) | 36.51 (28.04) | |
| | Ph.D. or equivalent | 11 (1.4%) | 30.68 (29.77) | |
| Smoking Habits | Non-smokers | 511 (64.8%) | 30.79 (22.73) | F = 1.75 P = 0.187 |
| | Smokers | 277 (35.2%) | 28.56 (22.44) | |
| Contact Lens (CL) use | Never | 648 (82.2%) | 27.97 (21.98) | F = 15.26 **P < 0.001** |
| | Sometimes | 102 (12.9%) | 39.24 (23.13) | |
| | Always | 38 (4.8%) | 39.91 (24.19) | |
| History of DED | No | 483 (61.3%) | 22.52 (18.90) | F = 50.14 **P < 0.001** |
| | Yes | 305 (38.7%) | 41.86 (23.03) | |
| OSDI Status | Normal | 203 (25.8%) | | |
| | Mild-to-Moderate | 170 (21.6%) | | |
| | Severe | 415 (52.7%) | | |

differences (P < 0.05) compared to other groups. No significant differences (P < 0.05) were observed among participants based on smoking status, field of study, or year of study.

### 3.2 Dry eye symptoms and tobacco use

Tobacco-related factors such as types of tobacco, rate of smoking, and areas of smoking were studied as shown in Table 2. Cigarette smokers (regular and electronic) demonstrated a comparable incidence of DE symptoms (OSDI score ≥ 13) and severity of symptoms (represented by mean OSDI scores) to the non-smokers. On the other hand, the highest incidence of DE symptoms (OSDI scores ≥ 13: 100% and 90%) and severity of symptoms (Mean OSDI: 48.13 and 38.08) were recorded among the smokers of Dokha and Ajami waterpipe, respectively.

**Table 2. OSDI status, mean scores (standard deviation), One-Way ANOVA, and distribution of responses (relative percentages) with respect to smoking types, rates, and areas.**

| Variable | | Normal | Mild-to-Moderate DE symptoms | Severe DE symptoms | Mean OSDI score (SD) |
|---|---|---|---|---|---|
| **Smoking Types** | **Regular Cigarette** | (35) 30.7% | (44) 38.6% | (35) 30.7% | 25.23 (21.41) |
| | **Electronic Cigarette (vape)** | (12) 17.4% | (31) 44.9% | (26) 37.7% | 30.65 (22.66) |
| | **Dokha (Medwakh pipe)** | (1) 10% | (2) 30% | (6) 60% | 48.13 (31.98) |
| | **Waterpipe (Mouassal)** | (20) 26.7% | (28) 37.3% | (27) 36% | 36.81 (20.85) |
| | **Waterpipe (Ajami)** | (0) 0% | (4) 44.4% | (5) 55.6% | 38.08 (21.29) |
| | **Non-smoker** | (111) 21.7% | (198) 38.7% | (202) 39.5% | 29.46 (22.69) |
| | One-Way ANOVA: F = 2.73, P = 0.019 | | | | |
| **Smoking Rate** | **Daily** | (4) 2.7% | (67) 45.9% | (75) 51.4% | 36.67 (21.04) |
| | **Weekly** | (9) 17.6% | (21) 41.2% | (21) 41.2% | 32.31 (23.53) |
| | **Monthly** | (42) 52.5% | (19) 23.8% | (19) 23.8% | 19.84 (20.61) |
| | **Non-smoker** | (124) 21.7% | (201) 38.7% | (186) 39.5% | 29.46 (22.69) |
| | One-Way ANOVA: F = 10.21, P < 0.001 | | | | |
| **Smoking Areas** | **Indoor** | (6) 13.3% | (15) 33.3% | (24) 53.3% | 36.94 (23.78) |
| | **Outdoor** | (17) 48.6% | (15) 42.9% | (3) 8.6% | 15.54 (18.69) |
| | **Mixed** | (39) 19.8% | (79) 40.1% | (79) 40.1% | 30.70 (21.92) |
| | **Non-Smoker** | (117) 21.7% | (199) 38.7% | (195) 39.5% | 29.46 (22.69) |
| | One-Way ANOVA: F = 6.37, P value < 0.001 | | | | |

A multiple regression analysis was conducted to examine the relationship between the number of smoking years, daily electronic device usage, daily study hours, and the OSDI score. The results are presented in Table 3.

Smoking rate (daily, weekly, monthly) showed a graded relationship with DE symptoms (incidence and severity). Expectedly, the highest incidence of DE symptoms (OSDI scores ≥ 13: 97.3%) and severity of symptoms (Mean OSDI: 36.67) were recorded among the daily smokers.

In addition, smoking environments (indoor, outdoor, or mixed) demonstrated a consistent association with the incidence and severity of symptoms. Expectedly, the highest incidence of DE symptoms (OSDI scores ≥ 13: 86.7%) and severity of symptoms (Mean OSDI: 36.94) were recorded among indoor smokers where bad ventilation areas are highly expected.

The One-Way ANOVA test showed significant differences among DE symptom severity (represented by OSDI mean scores) and all the Tobacco-related factors (P < 0.05) as represented in Table 2.

The OSDI score is evaluated as a function of three predictors: smoking years, electronic device usage hours per day, and the duration of studying hours per day. The regression equation derived from the unstandardized coefficients is as follows:

$$OSDI = 2.380 + 2.405(Smoking\ Years) + 1.668(Electronic\ Device\ Usage/Day) + 1.167(Duration\ of\ Studying\ Hours/Day)$$

The constant (B = 2.380) represents the predicted OSDI score when all predictors are set to zero. Smoking years (B = 2.405) have the largest unstandardized coefficient, indicating that each additional year of smoking is associated with a 2.405-point increase in the OSDI score, holding all other factors constant. Similarly, for every additional hour of electronic device usage, the OSDI score increases by 1.668, while an extra hour of studying per day results in a 1.167-point increase in the OSDI score. These findings highlight that all three predictors positively influence the OSDI score, with smoking years exerting the strongest impact.

The model exhibits a correlation coefficient (R = 0.600), indicating a moderate positive linear relationship between the predictors and the OSDI score. The coefficient of determination ($R^2$=0.360) reveals that 36.0% of the variance in the OSDI

**Table 3. Multi-regression analysis results.**

| Model Summary | | | | | | | |
|---|---|---|---|---|---|---|---|
| R | 0.600 | R-Square | 0.360 | | Adjusted R Square | | 0.353 |
| *ANOVA Test* | | F-value | 50.584 | | p-value | | <0.001 |
| *Stepwise model* | Coefficients | | | | Correlations | | |
| | Unstandardized | Standardized | t-value | Sig. | Zero-order | Partial | Part |
| Constant | 2.380 | --- | 0.870 | 0.385 | --- | --- | --- |
| Electronic device usage (hrs/day) | 2.405 | 0.432 | 8.141 | <0.001 | 0.534 | 0.444 | 0.396 |
| Duration of studying (hrs/day) | 1.668 | 0.239 | 4.471 | <0.001 | 0.430 | 0.263 | 0.218 |
| Smoking years | 0.167 | 0.134 | 2.732 | 0.007 | 0.195 | 0.164 | 0.133 |

score is explained by the predictors in the model. The adjusted $R^2$=0.353 further confirms this proportion after accounting for the number of predictors in the model. The ANOVA results, with an F-value of 50.584 and a highly significant p<0.001, indicate that the overall model is statistically significant, meaning that the predictor variables collectively have a meaningful impact on the OSDI score.

When considering the standardized coefficients (β), which allow for comparisons across variables on the same scale, smoking years remain the most influential predictor (β=0.432), followed by electronic device usage (β=0.239) and studying hours (β=0.134). These standardized coefficients indicate that smoking years have the strongest relative impact on the OSDI score, with electronic device usage and studying hours contributing moderately and weakly, respectively. This relative importance suggests that while all three variables are significant contributors, smoking years are the dominant driver of changes in the OSDI score.

The t-values and p-values further confirm the significance of these predictors. Smoking years exhibit a t-value of 8.141 (p<0.001), indicating a highly significant relationship with the OSDI score. Electronic device usage shows a t-value of 4.471 (p<0.001), affirming its moderate contribution. Studying hours, while weaker in effect, also demonstrate statistical significance with a t-value of 2.732 (p=0.007). These results establish that each predictor independently and significantly influences the OSDI score, with smoking years standing out as the most impactful.

The correlation values provide additional insights into the relationships between the predictors and the OSDI score. Smoking years display a strong zero-order correlation (r=0.534), reflecting a robust positive relationship with the OSDI score. Even after controlling the other predictors, the partial correlation (r=0.444) remains strong, and the part correlation (r=0.396) shows that smoking years uniquely account for approximately 15.7% of the variance in the OSDI score ($r^2$=$0.396^2$). Electronic device usage exhibits a moderate zero-order correlation (r=0.430) and a partial correlation (r=0.281) after adjusting for other variables, with a part correlation (r=0.238) indicating a unique contribution explaining about 5.7% of the variance. In contrast, the duration of studying hours per day shows a weak zero-order correlation (r=0.195), with a partial correlation (r=0.164) and a part correlation (r=0.133), uniquely explaining about 1.8% of the variance in the OSDI score. These findings emphasize the varying strengths of the predictors, with smoking years maintaining the strongest and most consistent influence.

In conclusion, this regression model demonstrates that smoking years, electronic device usage, and studying hours are significant predictors of the OSDI score, though their relative importance varies. Smoking years emerge as the most influential factor, followed by electronic device usage, while studying hours play comparatively minor role in explaining the variability in the score. The standardized coefficients and correlations provide a clear understanding of each variable's relative importance and unique contribution, highlighting the independent and combined effects of these predictors on the OSDI score.

**Analysis of study hours and electronic device usage across DE symptom severity levels.** A detailed examination of the distributions of electronic device usage and study hours across different DE symptom severity groups reveal

nuanced patterns that help explain the relatively weak associations observed in correlation analyses. While certain trends are apparent, the data does not support a straightforward or strongly linear relationship between these variables and severity of symptoms.

The first histogram illustrates the distribution of daily electronic device usage hours stratified by DE symptom severity, Fig 1. In the *Normal* group, the distribution is right-skewed, with the highest frequency of usage concentrated around 2–3 hours per day, and most individuals reporting between 1 and 7 hours of usage. The *Mild-to-Moderate* group demonstrates a higher central tendency, peaking at approximately 6 hours of daily use, with a noticeable spike involving around 65 participants. In contrast, the *Severe* group presents a multimodal distribution with prominent peaks at around 9 and 11 hours per day, suggesting higher average usage but with significant intra-group variability.

The second histogram captures the distribution of daily study hours across the same DE symptom severity categories, Fig 2. The *Normal* group displays a relatively symmetric distribution centered around 4–5 hours per day, with most participants studying between 2 and 8 hours. The *Mild-to-Moderate* group exhibits a slightly right-skewed pattern, with peak study duration at approximately 5–6 hours. The *Severe* group shows a broader distribution with a peak at around 7 hours, extending from approximately 4–10 hours per day.

Collectively, these findings suggest that while electronic device usage and study hours may be contributing factors, they do not independently account for substantial variation in DE symptom severity. The weak associations likely reflect the multifactorial nature of DE symptoms, in which simple exposure metrics are moderated by other critical influences such as screen brightness, viewing distance, blink rate, ambient humidity, and individual susceptibility. As such, future research should consider these additional variables to better elucidate the complex etiology of DE symptoms.

### 3.3 Dry eye symptoms and eye cosmetics usage

The use of eye cosmetics is a common tradition in the Arabic countries where females apply some of these products in proximity to the ocular surface. The current study aimed to investigate usage rates of several eye cosmetics and identify their association with DE symptoms (incidence and severity). The results are summarized in Table 4.

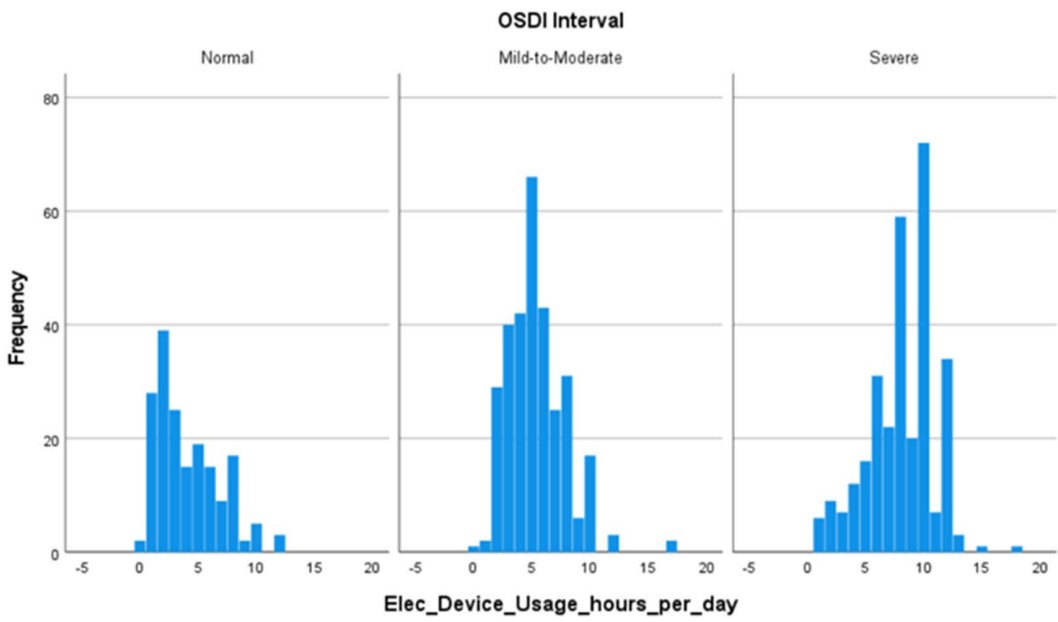

**Fig 1. Distribution of daily electronic device usage hours across the three DE symptom stages.**

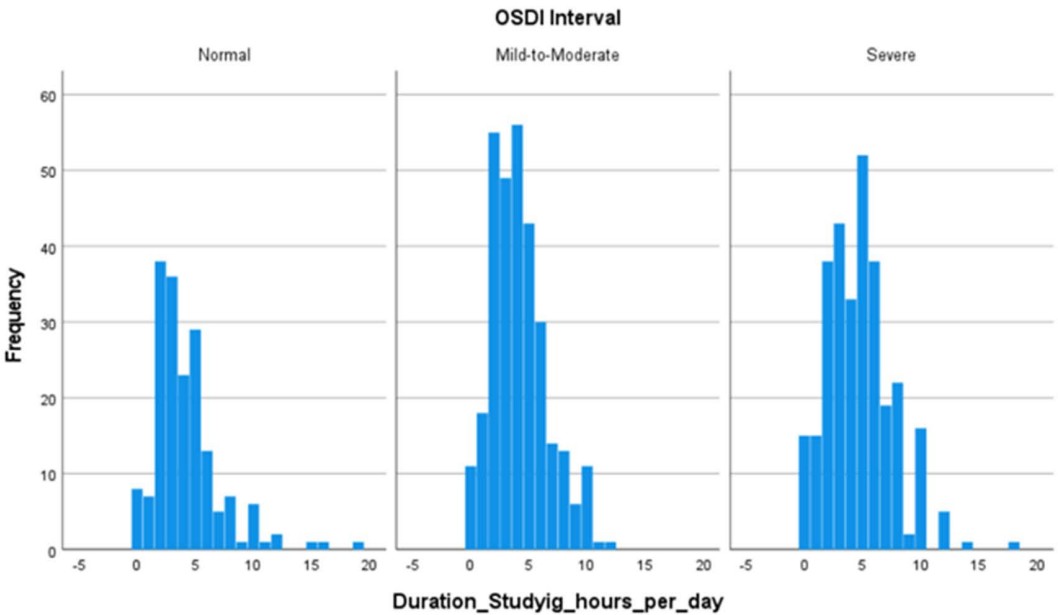

**Fig 2. Distribution of daily study hours across the three DE symptom stages.**

All types of eye cosmetics revealed similar DE symptoms status with respect to the usage rate. Expectedly, daily users of inner and external eyeliner demonstrated the highest incidence of DE symptoms (OSDI score ≥ 13: 89.4% and 90.7%) and DE symptom severity (Mean OSDI: 44.53 and 40.17), respectively. Also, daily users of false eyelashes, eye shadow and mascara demonstrated the highest incidence of DE symptoms (OSDI score ≥ 13: 89.5%, 87.2% and 92.4%) and DE symptom severity (Mean OSDI: 45.04, 38.24 and 44.16), respectively. The One-Way ANOVA test showed significant differences among DE symptom severity (represented by OSDI mean scores) and all the above-mentioned types of cosmetics (P < 0.05) as represented in Table 4.

Both duration of cosmetics usage factors (per year and per day) revealed a better graded relation with the DE symptoms (incidence and severity) as represented in Table 4. Expectedly, the highest incidence of DE symptoms (OSDI scores ≥ 13: 85.2% and 85.3%) and OSDI severity (mean OSDI: 37.78 and 39.28) were recorded among the users who wore eye cosmetics for more than 12 hours per day and 12 months per year, respectively. Focusing on the cleansing technique and rate, the highest incidence of DE symptoms (OSDI scores ≥ 13: 90.2% and 86.4%) and OSDI severity (mean OSDI: 40.78 and 38.41) were recorded among the users who used soap water for removing eye make and who rarely used cleansing cream, respectively. Finally, the One-Way ANOVA test showed considerable statistical significance among DE symptom severity (represented by OSDI mean scores) and all the factors of eye cosmetic use (p < 0.05) as represented in Table 5.

## Discussion

This study is the first study that estimated the incidence of DE symptoms and its association to behavioral-cultural factors such as tobacco and cosmetic use among the university students in Jordan. The study highlights the increasing severity of symptoms among young generations and underscores the need for awareness campaigns to mitigate its impact on quality of life. Additionally, the findings of the current study may encourage the development of new studies and public health initiatives, helping to design community-level awareness campaigns on DED and its contributing factors.

**Table 4. OSDI status, mean scores (standard deviation), One-Way ANOVA, and distribution of responses (relative percentages) with respect to usage rate of different types of eye cosmetics.**

| Variable | | Normal | Mild-to-Moderate DE Symptoms | Severe DE Symptoms | Mean OSDI score (SD) |
|---|---|---|---|---|---|
| **Inner Eyeliner** | **Daily** | 10.6% | 40.9% | 48.5% | 44.53 (28.01) |
| | **1-2 times per week** | 12.3% | 27.7% | 60% | 36.52 (23.95) |
| | **1-2 times per month** | 21.4% | 33% | 45.5% | 33.07 (22.91) |
| | **Never/NA** | 26.7% | 41.5% | 32.8% | 26.48 (21.74) |
| | One-Way ANOVA: F = 10.17, P < 0.001 | | | | |
| **External Eyeliner** | **Daily** | 9.3% | 33.3% | 57.3% | 40.17 (24.45) |
| | **1-2 times per week** | 20.4% | 35.7% | 43.8% | 38.21 (26.34) |
| | **1-2 times per month** | 15.6% | 36.5% | 47.9% | 34.35 (22.03) |
| | **Never/NA** | 26.4% | 41% | 32.6% | 26.51 (20.94) |
| | One-Way ANOVA: F = 10.47, P < 0.001 | | | | |
| **False Eyelashes** | **Daily** | 11.5% | 30.7% | 57.7% | 45.04 (24.79) |
| | **1-2 times per week** | 19% | 9.5% | 71.4% | 43.98 (34.98) |
| | **1-2 times per month** | 17.7% | 25.8% | 56.5% | 38.07 (24.39) |
| | **Never/NA** | 23.7% | 41.5% | 34.8% | 28.41 (21.94) |
| | One-Way ANOVA: F = 6.75, P < 0.001 | | | | |
| **Eye shadow** | **Daily** | 12.8% | 28.5% | 48.7% | 38.24 (20.59) |
| | **1-2 times per week** | 20.2% | 34.5% | 44.9% | 32.75 (25.03) |
| | **1-2 times per month** | 16.7% | 32.5% | 50.8% | 35.71 (22.32) |
| | **Never/NA** | 26.3% | 41.6% | 32.1% | 35.61 (22.79) |
| | One-Way ANOVA: F = 6.89, P < 0.001 | | | | |
| **Mascara** | **Daily** | 7.6% | 27.3% | 65.3% | 44.16 (23.75) |
| | **1-2 times per week** | 9.9% | 30.8% | 59.3% | 39.03 (22.23) |
| | **1-2 times per month** | 20.5% | 37.9% | 41.6% | 34.14 (23.67) |
| | **Never/NA** | 29.6% | 43.8% | 26.5% | 23.72 (19.82) |
| | One-Way ANOVA: F = 20.58, P < 0.001 | | | | |

The current study found that 74.2% of university students reported symptoms of DE based on an OSDI score >13. This proportion is comparable to previous studies conducted in Jordan, where the incidence of DE symptoms ranged from 59% to 73%, depending on the OSDI cut-off used [4,10].

In the current study, the high symptom burden among students may be attributed to factors such as limited awareness of DE symptoms, frequent contact lens use, tobacco exposure, cultural use of eye makeup, extensive digital device use, and environmental influences (e.g., air conditioning, desert dust, and UV exposure), all of which warrant further investigation.

The rising incidence of DE symptoms in Jordan could be attributed to the poor DED awareness among the university students, high proportions of contact lens use, tobacco exposure, cultural use of eye makeup, shifting of behavior towards the extensive use of digital screens as well as some environmental factors (e.g., air conditioning, UV sunlight, exposure to desert sand) [4,10].

Numerous studies have examined the smoking behaviors and cultural practices of university students in Jordan, reporting a prevalence of smoking (including cigarettes, pipes, and waterpipes) ranging from 28% to 42.7% [17–20]. In the current study, tobacco products such as Mouassal waterpipe, Ajami waterpipe, Medwakh, regular and e-cigarettes were investigated. The findings showed that Ajami water pipes smokers had recorded higher severity of DE symptoms than cigarettes which could be explained by the presence of higher quantity of irritants (i.e., Nicotine, Heavy metals, Carbon

**Table 5. OSDI status, mean scores (standard deviation), One-Way ANOVA, and distribution of responses (relative percentages) with respect to duration of cosmetic use (per day and per year); cleansing technique and rate; and sleep routine while wearing cosmetics.**

| Variable | | Normal | Mild-to-Moderate DE Symptoms | Severe DE Symptoms | Mean OSDI score (SD) |
|---|---|---|---|---|---|
| **Duration of Cosmetics use (per year)** | **< 6 months** | 19.5% | 37.6% | 42.9% | 33.07 (23.60) |
| | **6-12 months** | 18.9% | 26.4% | 54.7% | 36.32 (23.33) |
| | **>12 months** | 14.8% | 34.1% | 51.1% | 37.78 (24.11) |
| | **Don't use or NA** | 26.2% | 42.3% | 31.4% | 26.28 (21.04) |
| | One-Way ANOVA: F = 10.45, P < 0.001 | | | | |
| **Duration of Cosmetics use (per day)** | **< 6 hours** | 18.2% | 35.8% | 46% | 34.09 (23.34) |
| | **6-12 hours** | 18.4% | 34.4% | 47.2% | 34.43 (23.49) |
| | **>12 hours** | 14.3% | 33.3% | 52.4% | 39.28 (23.92) |
| | **Don't use or NA** | 26.7% | 42.4% | 30.9% | 26.07 (21.18) |
| | One-Way ANOVA: F = 10.73, P < 0.001 | | | | |
| **Cleansing Technique** | **Cleansing Cream** | 18.5% | 39.5% | 41.7% | 31.82 (21.81) |
| | **Soap Water** | 9.8% | 34.3% | 55.9% | 40.78 (25.31) |
| | **Water ONLY** | 19.2% | 31.3% | 49.5% | 34.68 (22.37) |
| | **Don't use or NA** | 30.4% | 42.6% | 27% | 23.89 (19.98) |
| | One-Way ANOVA: F = 23.59, P < 0.001 | | | | |
| **Cleansing Rate (per day)** | **Rarely** | 13.6% | 30.1% | 56.3% | 38.41 (23.63) |
| | **Sometimes** | 14.5% | 34.9% | 50.6% | 36.42 (23.24) |
| | **Always** | 17.2% | 38.5% | 44.4% | 33.87 (23.38) |
| | **Don't use or NA** | 28.6% | 42.3% | 29.1% | 25.26 (20.82) |
| | One-Way ANOVA: F = 15.76, P < 0.001 | | | | |
| **Sleep Routine** | **Sleeping while wearing eye cosmetics** | 13.6% | 38.6% | 47.7% | 38.25 (25.07) |
| | **Sleeping while wearing eye cosmetics and lenses** | 22% | 0% | 77.8% | 43.65 (22.56) |
| | **no lenses and eye cosmetics wear while sleeping or NA** | 23.9% | 40.4% | 35.7% | 28.72 (22.11) |
| | One-Way ANOVA: F = 8.59, P < 0.001 | | | | |

monoxide (CO), Polycyclic aromatic hydrocarbons (PAHs)) as well as the generation of more aerosols (puff smoke) in the waterpipes [21–23]. Among waterpipes, Mouassal tobacco may produce 30–40% less CO when mixed with molasses than Ajami waterpipe [24]. The findings of the current study also showed that Midwakh, which is a small pipe that filled with *dokha* (a blend of shredded tobacco leaves with a variety of barks, herbs, spices, dried flowers or dried fruit) [25], had recorded higher severity of DE symptoms than cigarettes. According to the literature, Midwakh is known its high nicotine content, containing up to five times as much as a regular cigarette and its smoking may release a mixture of toxic chemicals including nicotine, carbon monoxide, and oxidizing gases [25,26]. However, limited comparative information exposures between dokha and other tobacco products.

In addition to the harmful constituents of tobacco (including nicotine and toxic chemicals) which may permeate the bloodstream, affecting ocular tissues [27], the aerosol of puff smoke (gas-tar phases) are found to be also irritating to the ocular health where it may potentially increase the risk of lipid peroxidation of the outer layer of the tear film, and consequently contributing to the development or worsening of dry eye symptoms. [28]. Thus, the frequent smoking of tobacco as well as the indoor condition of smoking have found to be statistically associated with the incidence and severity of dry

eye symptoms among the study population which could be explained by the cumulative effect of aerosol irritants in poor ventilation conditions [29].

Similar to the effects of smoking, the use of eye cosmetics demonstrated a significant association with the incidence and severity of DE symptoms among the study population. Daily users of internal eyeliner and mascara exhibited the highest levels of DE symptom incidence and severity. This may be explained by the proximity of these cosmetics to the eyelid margin and their potential migration, leading to accidental exposure to irritative chemicals. Such exposure could contribute to tear film instability and/or inflammation of the corneal epithelium [30]. The frequent and long-term use of eye cosmetics along with poor hygiene practices would facilitate the accumulation of formulations at the lipid-aqueous interface of the tear film to reduce its stability and cause eye dryness [31]. In addition to the chemical-based irritants, biological-based irritants such as Demodex mites may persist in water-based formulations, potentially causing tear film irritation [32].

Eye hygiene plays an important role in decreasing the severity of DE symptoms among cosmetic users. As most cosmetic formulations are made of oily substances, cleansing cream (waterproof; slightly basic pH ≈ 7.3–7.4) was found to effectively remove eye makeup and reduce the severity of DE symptoms among the study population [32]. Soapy water was found to be less effective in cleaning and more associated with DE symptom severity likely due to the chemical components and alkaline nature of soap (basic pH values ≥ 8) [30,33].

This study has several limitations that may have contributed to the higher incidence of DE symptoms observed compared to previous studies. Data collection was based on an online self-administered questionnaire without accompanying clinical diagnostic tests, which may introduce response bias or social desirability effects, potentially influencing the findings and may have led to an overestimation of DE symptoms [34]. Furthermore, the study sample consisted predominantly of young university students, whose lifestyle and screen exposure patterns may differ significantly from those of the general population, potentially limiting the generalizability of the findings. Therefore, further research is recommended to clinically assess the incidence and severity of DE symptoms and explore its statistical association with tobacco and cosmetic use.

## Conclusion

Nearly three-quarters of university students in Jordan reported symptoms of DE as assessed by the validated Arabic version of the Ocular Surface Disease Index (OSDI). Symptoms of DE were significantly associated with female gender, older age (≥ 27 years), contact lens use, smoking, and eye makeup usage (P < 0.05). Among tobacco users, those who used Dokha or Ajami, smoked indoors, or smoked daily were more likely to report DE symptoms. Similarly, among cosmetic users, long-term use of mascara and internal eyeliner, especially when combined with poor hygiene practices, was associated with a higher likelihood of DE symptoms. These findings provide important baseline data for future research investigating behavioral and cultural risk factors for DED in the Jordanian population.

## Supporting information

**S1 File. The study questionnaire.**
(DOCX)

**S2 File. The coded data of the study.**
(XLSX)

## Author contributions

**Conceptualization:** May M. Bakkar, Mona Aridi, Wissam Ghach.

**Data curation:** May M. Bakkar, Mona Aridi, Mohammad A Alebrahim, Wissam Ghach.

**Formal analysis:** Mona Aridi, Wissam Ghach.

**Methodology:** May M. Bakkar, Mona Aridi, Wissam Ghach.

**Project administration:** May M. Bakkar, Mohammad A Alebrahim, Wissam Ghach.

**Supervision:** May M. Bakkar, Wissam Ghach.

**Writing – original draft:** May M. Bakkar, Wissam Ghach.

**Writing – review & editing:** May M. Bakkar, Mona Aridi, Mohammad A Alebrahim, Wissam Ghach.

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
