## [Decision Letter · Decision Letter 0]

17 Apr 2025

PONE-D-25-09013Prevalence of Symptomatic Dry Eye Disease and Behavioural-Cultural Risk Factors among University Students population in JordanPLOS ONE

Dear Dr. Ghach,

Thank you for submitting your manuscript to PLOS ONE. After careful consideration, we feel that it has merit but does not fully meet PLOS ONE’s publication criteria as it currently stands. Therefore, we invite you to submit a revised version of the manuscript that addresses the points raised during the review process. Namely, the reviewers have both asked to more accurately describe the actual study findings- prevalence of symptoms of dry eye, rather than clinical dry eye. In addition, the authors should be more careful in their wording about the results and implications of their findings.  

We look forward to receiving your revised manuscript.

Kind regards,

Liat Gantz, PhD

Academic Editor

PLOS ONE

 [Research Grant No: 20230271 from Deanship of Research at Jordan University of Science and Technology]. 

Additional Editor Comments:

The manuscript has been reviewed by two external reviewers who agree that this manuscript should be revised to more accurately state what the authors have examined, symptoms of dry eye rather than clinical diagnosis of dry eye.

One reviewer asked to re-analyze based on an OSDI score greater than 20. I would ask that you specify this in addition to the >13 cut off.

The manuscript can be further improved based on additional comments below:

Abstract

Line 35 Google forms and not survey

The prevalence of symptoms of DED, not DED- as DED is not diagnosed just based on symptoms

Line 169: The word “Although” is inappropriately used in this sentence

Line 170: Were they not diagnosed with DE clinically or based only on the OSDI questionnaire, in which case the authors mean that 61.3% did not report symptoms of DE

Line 173: As seen in Table 1 instead of “according to Table 1”

What is “history of DED” in Table 1? Is it a self-report? Based on symptoms or an actual diagnosis?

The whole concept of DED severity based on OSDI score value vs. DED diagnosis based on OSDI>13 is unfounded (such as in Line 199). One could say that the population with OSDI scores > 13 are considered having symptoms of DE and higher scores indicate higher symptomology. However, no claim can be made about DE severity or the existence of DE.

Line 193: what is harmonic relationship

Line 197: Also? In any case better replaced with “In addition,”

Table 2- include numbers and not just percentages

Don’t multiple regressions require a different p-value that takes into account the multiple calculations (Bonferroni corrected)?

Lines 263-272: should be rephrased to emphasize the minor role of study hours accounting for under 2% of the score rather than to state several times that this is a predictor with a unique contribution.

Line 291: what does harmonic mean?

Line 295: wore instead of had utilized

Line 298: used instead of had utilized

Line 308: estimated the prevalence of symptoms of DE amongst.. not the prevalence of symptomatic DED as no clinical measure was tested

Lines 312-315: very far fetched, suggest removing: “The anticipated outcome is to provide ophthalmologists and eye health researchers with valuable new evidence on the prevalence and risk factors of DED in Jordanian universities.”

Discussion, second paragraph: The previous study reporting 59% of non clinical populations with symptoms of DE used a cut-off of OSDI>20 and was careful not to state prevalence of DE but symptoms of DE.

The other study used only OSDI and considered a score >13 as the only determinant of DE (which is incorrect) and found a similar prevalence of symptoms of DE to the present study.

This should be emphasized in the discussion.

Line 366: Cleansing creams should be the beginning of a new sentence

Line 369: less effective in cleaning

Line 370: “due to”- you have no proof what this is due to. Please rephrase. “This could be due to the chemical components and characteristics.. as shown in XXX”.

Modify conclusions based on what was examined- symptoms only:

Nearly three-quarters of university students in Jordan exhibited symptoms of DED as assessed by the validated Arabic version of the Ocular Surface Disease Index (OSDI). Symptoms of DED were significantly positively associated with female gender, older age (≥27 years), contact lens use, smoking, and eye makeup usage. Etc.

Reviewers' comments:

Reviewer's Responses to Questions

**Comments to the Author**

1. Is the manuscript technically sound, and do the data support the conclusions?

Reviewer #1: Yes

Reviewer #2: Yes

2. Has the statistical analysis been performed appropriately and rigorously? 

Reviewer #1: Yes

Reviewer #2: Yes

3. Have the authors made all data underlying the findings in their manuscript fully available?

Reviewer #1: Yes

Reviewer #2: No

4. Is the manuscript presented in an intelligible fashion and written in standard English?

Reviewer #1: Yes

Reviewer #2: Yes

5. Review Comments to the Author

Reviewer #1: This study assessed the prevalence of symptomatic dry eye disease (DED) among university students in Jordan and its association with behavioral and cultural risk factors. Defined by an OSDI score ≥ 13, DED prevalence was 74.2%. They concluded that symptomatic DED is highly prevalent among university students in Jordan and is significantly associated with factors such as age, gender, contact lens use, cosmetic application, and tobacco consumption.

The manuscript contains important findings; however, it needs to undergo revision to enhance its significance.

Abstract:

* The research objective needs to be more precise. You stated:

"To estimate the prevalence of symptomatic dry eye disease (DED) among university students in Jordan and examine the relationship between behavioral and cultural risk factors and the severity of DED."

However, you did not assess the prevalence of dry eye disease, as you did not diagnose DED according to standard criteria (OSDI symptom questionnaire≥ 13 plus at least one positive clinical test—NITBUT, osmolarity, or staining), as recommended by TFOS. Instead, you assessed the prevalence of dry eye symptoms, not the prevalence of individuals with dry eye disease who exhibit symptoms.

*Rephrasing the objective and conclusions is necessary, especially considering that 61.3% of the participants had not been clinically diagnosed with DED.

*The results section of the abstract should include the mean age of the participants, standard deviation, and age range.

*It is standard to include P-values and significant results within the results section of the abstract.

Introduction:

On page 3, lines 67-68, it is stated that the prevalence of dry eye disease in Jordan is 59%. How was dry eye diagnosed? Was it through a questionnaire alone, or were clinical signs also used?

Methods:

*The statistical analysis lacks a sample size calculation.

*It is not stated whether a distribution test was performed on the data. Is the distribution normal or not?

Results:

*The mean age of the participants and standard deviation should be added. If the age distribution is not normal, the median and quartiles (Q1, Q3) should also be included.

Discussion:

*It should be clarified that you assessed the prevalence of dry eye symptoms and not the prevalence of individuals with dry eye disease who are symptomatic (page 15, lines 308-310).

*It is recommended to discuss potential biases or methodological limitations that could account for the higher prevalence observed in this study compared to previous research.

Reviewer #2: This represents an interesting study regarding the prevalence of dry eye symptoms.

What instructions were given to participants? The OSDI questionnaire contains questions referring to blurred vision that may be confused with uncorrected refractive error.

Why was the OSDI Cutoff Score of 13 used, rather than 20 as in previous study? (Bakkar et al Epidemiology of symptoms of dry eye disease (DED) in Jordan: Across-sectional non-clinical population-based study. Contact Lens and Anterior Eye 39 (2016) 197–202). For better comparison with the general population in Jordan it would make sense to use the same cutoff score. Especially, as a score of 13 represents very mild symptoms.

Line 68: the study only explored symptoms of DED, please adjust accordingly.

Line 169: please delete ‘although’ – Does this mean that 38.7 % of the participants have been previously diagnosed with DED? Please explain. If indeed, this large proportion of participants already have a DED diagnosis, a sub analysis of this group in view of risk factors may be warranted in view of the large sample size in this study.

Results:

In view of the very large standard deviations, it would be very interesting to include histograms similar to Figures 1-3 in your previous publication (Bakkar et al 2016).

6. PLOS authors have the option to publish the peer review history of their article (what does this mean? ). If published, this will include your full peer review and any attached files.

**Do you want your identity to be public for this peer review?** For information about this choice, including consent withdrawal, please see our Privacy Policy .

Reviewer #1: No

Reviewer #2: No

---

## [Author Response · Author response to Decision Letter 1]

13 May 2025

Response to Reviewers

We would like to sincerely thank the reviewers for their thorough evaluation of our manuscript and for providing valuable and insightful comments. We appreciate the time and effort invested in reviewing our work and recognize that the suggestions offered have significantly contributed to improving the clarity, rigor, and overall quality of the manuscript.

In the following document, we have addressed each comment carefully and made the necessary revisions to strengthen the study.

We hope that the changes meet the reviewers’ expectations and enhance the impact of our research.

Reviewer #1:

This study assessed the prevalence of symptomatic dry eye disease (DED) among university students in Jordan and its association with behavioral and cultural risk factors. Defined by an OSDI score ≥ 13, DED prevalence was 74.2%. They concluded that symptomatic DED is highly prevalent among university students in Jordan and is significantly associated with factors such as age, gender, contact lens use, cosmetic application, and tobacco consumption. The manuscript contains important findings; however, it needs to undergo revision to enhance its significance.

Abstract:

* The research objective needs to be more precise. You stated:

"To estimate the prevalence of symptomatic dry eye disease (DED) among university students in Jordan and examine the relationship between behavioral and cultural risk factors and the severity of DED."

However, you did not assess the prevalence of dry eye disease, as you did not diagnose DED according to standard criteria (OSDI symptom questionnaire≥ 13 plus at least one positive clinical test—NITBUT, osmolarity, or staining), as recommended by TFOS. Instead, you assessed the prevalence of dry eye symptoms, not the prevalence of individuals with dry eye disease who exhibit symptoms.

We acknowledge the distinction that the reviewer has highlighted between symptomatic dry eye and clinically diagnosed dry eye disease as defined by the TFOS DEWS II criteria. It is correct that our study utilized the OSDI questionnaire alone without accompanying clinical tests such as NITBUT, osmolarity, or ocular surface staining. Therefore, our findings pertain specifically to the prevalence of dry eye symptoms rather than confirmed dry eye disease. This is stated clearly in the manuscript title /aims: “Prevalence of Symptomatic Dry Eye Disease and Behavioural-Cultural Risk Factors among University Students population in Jordan”

In response to the reviewer’s comment, we will revise the objective in the manuscript to more accurately reflect this, as follows:

"To estimate the prevalence of dry eye symptoms among university students in Jordan and examine the relationship between behavioral and cultural risk factors and symptom severity. Additionally, the limitation of relying solely on symptoms without clinical confirmation of DED was acknowledged in the study limitations (lines 388-398)

*Rephrasing the objective and conclusions is necessary, especially considering that 61.3% of the participants had not been clinically diagnosed with DED.

Thank you for your insightful comment. We have revised both the objective and conclusions to clearly state that our results pertain to dry eye symptoms rather than clinically diagnosed dry eye disease. These revisions aim to ensure clarity and alignment with the data presented.

*The results section of the abstract should include the mean age of the participants, standard deviation, and age range.

*It is standard to include P-values and significant results within the results section of the abstract.

We sincerely thank the reviewer for this valuable comment. In response, we have revised the Results section of the abstract to include the mean age (21.87 years), standard deviation (3.824), and age range (18–45 years) of the study participants. Furthermore, we have incorporated the p-values corresponding to statistically significant findings to enhance the clarity and rigor of the results presentation. The revised abstract has been updated accordingly.

Introduction:

On page 3, lines 67-68, it is stated that the prevalence of dry eye disease in Jordan is 59%. How was dry eye diagnosed? Was it through a questionnaire alone, or were clinical signs also used?

Thank you for your comment and for pointing this out. The reported 59% prevalence of dry eye disease in Jordan, as cited on page 3 (lines 71-75), was based on symptoms only and assessed using a non-validated version of the OSDI questionnaire. No clinical signs or diagnostic tests were used in that previous study. We also note that this earlier report was co-authored by one of the current co-authors of this submission, which we acknowledge in the context of the literature cited.

To avoid any confusion, we will clarify this point in the manuscript lines 67-68 to ensure an accurate interpretation of the referenced prevalence figure.

Methods:

*The statistical analysis lacks a sample size calculation.

We acknowledge that a formal sample size calculation was not conducted prior to data collection, as the study adopted a convenience sampling method targeting university students across all Jordanian governorates. Our primary aim was to maximize participation by distributing the survey widely through social media platforms and university research departments, rather than limiting the sample to a predefined number.

Nonetheless, with a final sample of 788 participants, the study achieved a sufficiently large cohort to provide robust and generalizable findings for the target population. To strengthen the methodological transparency, we have now added a statement to the "Materials and Methods" section clarifying the sampling strategy and the absence of a priori sample size calculation.

*It is not stated whether a distribution test was performed on the data. Is the distribution normal or not?

ANOVA is known to be robust to violations of normality, particularly with large sample sizes. Given that the current sample size is 788, well above the commonly accepted thresholds (n > 30–40), the Central Limit Theorem ensures that the sampling distribution of the mean approximates normality. Therefore, the use of one-way ANOVA remains appropriate, and there was no need to switch to a nonparametric alternative [1].

Results:

*The mean age of the participants and standard deviation should be added. If the age distribution is not normal, the median and quartiles (Q1, Q3) should also be included.

We have revised the Results section to include the mean age and standard deviation of the participants.

The mean age of the participants was 21.87 years (SD = 3.824).

Discussion:

*It should be clarified that you assessed the prevalence of dry eye symptoms and not the prevalence of individuals with dry eye disease who are symptomatic (page 15, lines 308-310).

Thank you for your comment. We have revised the manuscript to clarify that our study assessed the prevalence of symptomatic dry eye rather than the prevalence of clinically diagnosed dry eye disease. The objective, results, discussion and conclusion sections have all been amended accordingly to accurately reflect this distinction. The text, has also been updated to ensure consistency with this clarification.

*It is recommended to discuss potential biases or methodological limitations that could account for the higher prevalence observed in this study compared to previous research.

We have expanded the discussion section to address potential biases and methodological limitations that may have contributed to the higher prevalence of dry eye symptoms observed in our study compared to previous research. Refer to Lines 392-401

Reviewer #2:

This represents an interesting study regarding the prevalence of dry eye symptoms.

What instructions were given to participants? The OSDI questionnaire contains questions referring to blurred vision that may be confused with uncorrected refractive error.

Why was the OSDI Cutoff Score of 13 used, rather than 20 as in previous study? (Bakkar et al Epidemiology of symptoms of dry eye disease (DED) in Jordan: Across-sectional non-clinical population-based study. Contact Lens and Anterior Eye 39 (2016) 197–202). For better comparison with the general population in Jordan it would make sense to use the same cutoff score. Especially, as a score of 13 represents very mild symptoms.

Thank you for your thoughtful feedback. Participants were instructed to complete the OSDI questionnaire based on their subjective experience of dry eye symptoms during the previous two weeks, without consideration of any known refractive error or visual correction. We acknowledge that certain questions, particularly those referring to blurred vision, may have been influenced by uncorrected refractive error, which could introduce some response bias.

Regarding the cutoff score, yes, a previous report by Bakkar et al. used a cutoff of 20; however, we relied on more recent studies that use a cutoff value of 13 to define symptomatic dry eye, as this has become a widely accepted threshold in current literature. Using this cutoff improves consistency with more recent research and facilitates broader comparisons. This rationale has been clarified in the methods and discussion sections of the manuscript.

Line 68: the study only explored symptoms of DED, please adjust accordingly.

Thank you for your comment. We have revised the manuscript to clarify that our study assessed the prevalence of symptomatic dry eye rather than the prevalence of clinically diagnosed dry eye disease. The objective, results, discussion and conclusion sections have all been amended accordingly to accurately reflect this distinction. The text, has also been updated to ensure consistency with this clarification.

Line 169: please delete ‘although’ – Does this mean that 38.7 % of the participants have been previously diagnosed with DED? Please explain. If indeed, this large proportion of participants already have a DED diagnosis, a sub analysis of this group in view of risk factors may be warranted in view of the large sample size in this study.

We have removed the word "although" for clarity. To clarify, 38.7% of the participants self-reported a previous diagnosis of DED by an eye care professional. However, we acknowledge that this was based on self-reporting and not verified through medical records or standardized clinical criteria, which may introduce recall or reporting bias.

Given your helpful suggestion and the size of the sample, we agree that a sub-analysis of this subgroup could provide additional insights into potential differences in risk factor profiles. We have now included this as a consideration for future research in the discussion section.

Results:

In view of the very large standard deviations, it would be very interesting to include histograms similar to Figures 1-3 in your previous publication (Bakkar et al 2016).

We have included two histograms in the revised manuscript (Figure1 and Figure 2) to illustrate the distribution of electronic device usage hours and study hours across different DED severity groups. These visualizations provide a clearer understanding of the observed variability and support the interpretation of the relatively weak associations between these predictors and DED severity.

[1] M. Blanca, R. Alarcón, J. Arnau, R. Bono, and R. Bendayan, ‘Non-normal data: Is ANOVA still a valid option?’, Psicothema, vol. 4, no. 29, pp. 552–557, Nov. 2017, doi: 10.7334/psicothema2016.383.

Response to Additional Comments from the Editor

We would like to sincerely thank the Editor for the careful evaluation of our manuscript and for overseeing the review process. We appreciate the constructive feedback and the opportunity to revise our work.

All comments and suggestions have been thoroughly addressed, and we believe the manuscript has been significantly improved as a result.

The manuscript has been reviewed by two external reviewers who agree that this manuscript should be revised to more accurately state what the authors have examined, symptoms of dry eye rather than clinical diagnosis of dry eye.

One reviewer asked to re-analyze based on an OSDI score greater than 20. I would ask that you specify this in addition to the >13 cut off.

Thank you for your comment. We have revised the manuscript to clarify that our study assessed the prevalence of symptomatic dry eye rather than the prevalence of clinically diagnosed dry eye disease. The objective, results, discussion and conclusion sections have all been amended accordingly to accurately reflect this distinction. The text, has also been updated to ensure consistency with this clarification.

Regarding the cutoff score, yes, a previous report by Bakkar et al. used a cutoff of 20; however, we relied on more recent studies that use a cutoff value of 13 to define symptomatic dry eye, as this has become a widely accepted threshold in current literature. Using this cutoff improves consistency with more recent research and facilitates broader comparisons. This rationale has been clarified in the methods and discussion sections of the manuscript.

The manuscript can be further improved based on additional comments below:

Abstract

Line 35 Google forms and not survey

Adjusted.

The prevalence of symptoms of DED, not DED- as DED is not diagnosed just based on symptoms

We have revised the manuscript to clarify that our study assessed the prevalence of symptomatic dry eye rather than the prevalence of clinically diagnosed dry eye disease.

Line 169: The word “Although” is inappropriately used in this sentence

The word “Although” is now replaced by “Additionally” which is more convenient.

Line 170: Were they not diagnosed with DE clinically or based only on the OSDI questionnaire, in which case the authors mean that 61.3% did not report symptoms of DE

The 61.3% figure refers to participants who did not report symptoms of dry eye disease (DE) based on their OSDI questionnaire scores. These participants were not clinically diagnosed with DE; rather, symptom assessment was based solely on the OSDI, as noted in the methodology.

Line 173: As seen in Table 1 instead of “according to Table 1”

Adjusted.

What is “history of DED” in Table 1? Is it a self-report? Based on symptoms or an actual diagnosis?

The whole concept of DED severity based on OSDI score value vs. DED diagnosis based on OSDI>13 is unfounded (such as in Line 199). One could say that the population with OSDI scores > 13 are considered having symptoms of DE and higher scores indicate higher symptomology. However, no claim can be made about DE severity or the existence of DE.

We thank the reviewer for this important observation. In Table 1, “history of DED” refers to participants’ self-reported history of a prior dry eye disease diagnosis by a healthcare professional. We agree with the reviewer that OSDI scores alone cannot establish a clinical diagnosis or severity of DED. Accordingly, we have revised the manuscript to clarify that OSDI scores >13 indicate the presence of dry eye symptoms, not a definitive diagnosis or clinical severity of DED. We have also modified the language in Line 199 and related sections to reflect this distinction more accurately.

Line 193: what is harmonic relationship

It means correlated. For much more accurate writing the word harmonic replaced with graded.

Line 197: Also? In any case better replaced with “In addition,”

Adjusted.

Table 2- include numbers and not just percentages

Done

Don’t multiple regressions require a different p-value that takes into account the multiple calculations (Bonferroni corrected)?

We appreciate the opportunity to clarify our approach. In the multiple regression analysis conducted, Bonferroni correction was not applied because the predictors were evaluated simultaneously within a single regression model rather than through multiple independent hypothesis tests.

In standard practice, the overall model significance (as indicated by the ANOVA test, p < 0.001) controls for Type I error at the model level. Furthermore, the individual p-values for the predictors assess the significance of each variable conditional on the others included in the model.

Th

---

## [Decision Letter · Decision Letter 1]

16 Jun 2025

PONE-D-25-09013R1Prevalence of Symptomatic Dry Eye Disease and Behavioural-Cultural Risk Factors among University Students population in JordanPLOS ONE

Dear Dr. Ghach,

Thank you for submitting your manuscript to PLOS ONE. After careful consideration, we feel that it has merit but does not fully meet PLOS ONE’s publication criteria as it currently stands. Therefore, we invite you to submit a revised version of the manuscript that addresses the points raised during the review process.Please ensure that your decision is justified on PLOS ONE’s publication criteria  and not, for example, on novelty or perceived impact.

Both reviewers feel that the revision addresses the comments. I still would like to see the authors refining their use of "symptomatic DED" which implies that the authors examined the presence of DED- which they did not, by replacing it with "dry eye symptoms". Similarly "severity of DED" should be replaced with "Severity of symptoms". Additional minor comments below as well.

We look forward to receiving your revised manuscript.

Kind regards,

Liat Gantz, PhD

Academic Editor

PLOS ONE

Journal Requirements:

Additional Editor Comments:

Both reviewers feel that the revision addresses the comments. I still would like to see the authors refining their use of "symptomatic DED" which implies that the authors examined the presence of DED- which they did not, by replacing it with "dry eye symptoms". Similarly "severity of DED" should be replaced with "Severity of symptoms". Additional minor comments below as well:

Abstract

Your study only assessed incidence of symptoms and not of symptomatic dry eye disease- therefore, you must correctly refer to your study as assessing dry eye symptoms

Page 3 Line 88: the prevalence of dry eye symptoms, not symptomatic DED

Page 4 Line 101: dry eye symptoms, not symptomatic DED

Page 4: please calculate the power of a sample with 788 respondents precisely and not “provided a sufficiently large database”

Page 8 Line 179: instead of “Regarding the participants’ field and year of study,” please rephrase: “Of the participants,”

Pages 9-10 Line 201, 203, 207, 208, 215: it is not DED severity, it is dry eye symptom severity

Tables 2, 4, 5: severity of SYMPTOMS, not of DED- you do not know the severity of DED as you did not measure it at all

Line 288, Line 293, Line 295, Line 304, Line 315, Line 326, Line 327m Line 329Line 331,Line 334, Line 341 (here it is stated as DED prevalence and severity, both were not examined), Line 342, Line 350,: It is NOT DED severity, it is Symptoms severity

Line 348: used instead of utilized

Page 19 Line 357: dry eye symptoms and not symptomatic DED

Line 359: severity of symptoms, not symptomatic DED

Line 378: dry eye symptoms

Line 389: dry eye symptoms

Line 423: dry eye symptom severity

Line 431: studies instead of research

Line 449: dry eye symptoms instead of symptomatic DED

Line 451: the Jordanian population

Reviewers' comments:

Reviewer's Responses to Questions

**Comments to the Author**

1. If the authors have adequately addressed your comments raised in a previous round of review and you feel that this manuscript is now acceptable for publication, you may indicate that here to bypass the “Comments to the Author” section, enter your conflict of interest statement in the “Confidential to Editor” section, and submit your "Accept" recommendation.

Reviewer #1: All comments have been addressed

Reviewer #2: All comments have been addressed

2. Is the manuscript technically sound, and do the data support the conclusions?

Reviewer #1: Yes

Reviewer #2: Yes

3. Has the statistical analysis been performed appropriately and rigorously? 

Reviewer #1: Yes

Reviewer #2: Yes

4. Have the authors made all data underlying the findings in their manuscript fully available?

Reviewer #1: Yes

Reviewer #2: Yes

5. Is the manuscript presented in an intelligible fashion and written in standard English?

Reviewer #1: Yes

Reviewer #2: Yes

6. Review Comments to the Author

Reviewer #1: The authors have addressed the suggested improvements, and therefore I believe the manuscript is ready for publication.

Reviewer #2: I am happy with the revisions, I feel that the authors have sufficientyl addressed my queries and I support this publication.

7. PLOS authors have the option to publish the peer review history of their article (what does this mean? ). If published, this will include your full peer review and any attached files.

**Do you want your identity to be public for this peer review?** For information about this choice, including consent withdrawal, please see our Privacy Policy .

Reviewer #1: No

Reviewer #2: **Yes: ** Daniela Sonja Nosch

---

## [Author Response · Author response to Decision Letter 2]

18 Jun 2025

Rebuttal letter

Both reviewers feel that the revision addresses the comments. I still would like to see the authors refining their use of "symptomatic DED" which implies that the authors examined the presence of DED- which they did not, by replacing it with "dry eye symptoms". Similarly "severity of DED" should be replaced with "Severity of symptoms". All the terms “prevalence”, “symptomatic DED”, and “DED severity” were replaced by “incidence”, “Dye eye symptoms”, and “severity of symptoms”, respectively

Additional minor comments below as well:

Abstract

Your study only assessed incidence of symptoms and not of symptomatic dry eye disease- therefore, you must correctly refer to your study as assessing dry eye symptoms

Page 3 Line 88: the prevalence of dry eye symptoms, not symptomatic DED(corrected)

Page 4 Line 101: dry eye symptoms, not symptomatic DED (corrected)

Page 4: please calculate the power of a sample with 788 respondents precisely and not “provided a sufficiently large database”

A post hoc power analysis was conducted using G*Power (version 3.1.9.7) for the one-way ANOVA tests applied in the study. The total sample consisted of 788 university students, and power calculations were performed for the two primary categorical independent variables used in the ANOVA models, which included variables with 4 groups (smoking rate, smoking areas, and all the cosmetics variables) and 6 groups (smoking type).

For the variable with 4 groups, the analysis yielded a non-centrality parameter (λ) = 49.25, numerator degrees of freedom (df) = 3, denominator df = 784, a critical F value = 2.616, and a statistical power (1 − β) = 0.999995.

Similarly, for the variable with 6 groups, the parameters were: λ = 49.25, numerator df = 5, denominator df = 782, critical F value = 2.226, and statistical power = 0.999977.

These results confirm that the achieved sample size was statistically robust and sufficiently powered to detect moderate effects across all ANOVA tests used in the study, thereby justifying the absence of an a priori sample size calculation.

This is now clearly addressed in the text in page 4, as follows:

“To further verify the adequacy of the sample size, a post hoc power analysis was conducted using G*Power (version 3.1.9.7) for the One-way ANOVA tests employed in the study. Power estimates were calculated for variables with four groups (smoking rate, smoking areas, and all the cosmetics variables) and 6 groups (smoking type), which represent the range of groupings used in the analysis. For the variable with four groups, the analysis yielded a non-centrality parameter (λ) = 49.25, numerator df = 3, denominator df = 784, critical F = 2.616, and a power of 0.999995. For the variable with six groups, the results were similarly robust: λ = 49.25, numerator df = 5, denominator df = 782, critical F = 2.226, and power = 0.999977. These results confirm that the sample size was statistically robust and sufficiently powered to detect moderate effects across the ANOVA models used in the study.”

Page 8 Line 179: instead of “Regarding the participants’ field and year of study,” please rephrase: “Of the participants,” (corrected)

Pages 9-10 Line 201, 203, 207, 208, 215: it is not DED severity, it is dry eye symptom severity

Tables 2, 4, 5: severity of SYMPTOMS, not of DED- you do not know the severity of DED as you did not measure it at all (corrected)

Line 288, Line 293, Line 295, Line 304, Line 315, Line 326, Line 327m Line 329Line 331,Line 334, Line 341 (here it is stated as DED prevalence and severity, both were not examined), Line 342, Line 350,: It is NOT DED severity, it is Symptoms severity (corrected)

Line 348: used instead of utilized (corrected)

Page 19 Line 357: dry eye symptoms and not symptomatic DED (corrected)

Line 359: severity of symptoms, not symptomatic DED (corrected)

Line 378: dry eye symptoms (corrected)

Line 389: dry eye symptoms (corrected)

Line 423: dry eye symptom severity (corrected)

Line 431: studies instead of research (corrected)

Line 449: dry eye symptoms instead of symptomatic DED (corrected)

Line 451: the Jordanian population (corrected)

---

## [Editor Report · Decision Letter 2]

20 Jun 2025

PONE-D-25-09013R2Incidence of Dry Eye Symptoms and Behavioural-Cultural Risk Factors among University Students population in Jordan

PLOS ONE

Dear Dr. Ghach,

Thank you for submitting your manuscript to PLOS ONE. After careful consideration, we feel that it has merit but does not fully meet PLOS ONE’s publication criteria as it currently stands. Therefore, we invite you to submit a revised version of the manuscript that addresses the points raised during the review process.

The authors should re-read their manuscript and refer to symptoms of dry eye when they are referring to their study which is only based on OSDI. However, in the introduction and discussion - in instances where the authors quote previous studies which examined dry eye disease using clinically acceptable diagnostic criteria- they should not refer to dry eye symptoms. The way the paper was submitted appears as if the authors replaced every single instance of the words "dry eye" with "dry eye symptoms (DES)" including places such as the TFOS guidelines for diagnosing dry eye disease (and not just symptoms).

Further DES- is often used for digital eye strain and even dry eye syndrome. Please refrain from the abbreviation and write DE symptoms.

The authors are requested to please seriously review and read their paper before resubmitting a manuscript in the current state.

We look forward to receiving your revised manuscript.

Kind regards,

Liat Gantz, PhD

Academic Editor

PLOS ONE

Journal Requirements:

Additional Editor Comments:

The authors should re-read their manuscript and refer to symptoms of dry eye when they are referring to their study which is only based on OSDI. However, in the introduction and discussion - in instances where the authors quote previous studies which examined dry eye disease using clinically acceptable diagnostic criteria- they should not refer to dry eye symptoms. The way the paper was submitted appears as if the authors replaced every single instance of the words "dry eye" with "dry eye symptoms (DES)" including places such as the TFOS guidelines for diagnosing dry eye disease (and not just symptoms).

Further DES- is often used for digital eye strain and even dry eye syndrome. Please refrain from the abbreviation and write DE symptoms.

The authors are requested to please seriously review and read their paper before resubmitting a manuscript in the current state.

---

## [Author Response · Author response to Decision Letter 3]

24 Jun 2025

The authors should re-read their manuscript and refer to symptoms of dry eye when they are referring to their study which is only based on OSDI. However, in the introduction and discussion - in instances where the authors quote previous studies which examined dry eye disease using clinically acceptable diagnostic criteria- they should not refer to dry eye symptoms. The way the paper was submitted appears as if the authors replaced every single instance of the words "dry eye" with "dry eye symptoms (DES)" including places such as the TFOS guidelines for diagnosing dry eye disease (and not just symptoms).

Further DES- is often used for digital eye strain and even dry eye syndrome. Please refrain from the abbreviation and write DE symptoms.

The authors are requested to please seriously review and read their paper before resubmitting a manuscript in the current state.

Answer: All the suggested comments by the editor were considered in the revised manuscript.

---

## [Editor Report · Decision Letter 3]

29 Jun 2025

Incidence of Dry Eye Symptoms and Behavioural-Cultural Risk Factors among University Students population in Jordan

PONE-D-25-09013R3

Dear Dr. Ghach,

We’re pleased to inform you that your manuscript has been judged scientifically suitable for publication and will be formally accepted for publication once it meets all outstanding technical requirements.

Kind regards,

Liat Gantz, PhD

Academic Editor

PLOS ONE

Additional Editor Comments (optional):

Dear authors- thank you for addressing the DED/DES issue stated earlier. I think it is ready for publication now. I encourage you to review the proofs carefully as I have noticed two instances of usage of "DED" instead of "DE symptoms". Good luck and thank you for your contribution.
---

## [Editor Report · Acceptance letter]

PONE-D-25-09013R3

PLOS ONE

Dear Dr. Ghach,

I'm pleased to inform you that your manuscript has been deemed suitable for publication in PLOS ONE. Congratulations! Your manuscript is now being handed over to our production team.

Kind regards,

on behalf of

Dr. Liat Gantz

Academic Editor

PLOS ONE